# Oncometabolic role of mitochondrial sirtuins in glioma patients

**Maria Fazal Ul Haq**[1☯], **Muhammad Zahid Hussain**[2], **Ishrat Mahjabeen**[1☯]*,
**Zertashia Akram**[1], **Nadia Saeed**[1], **Rabia Shafique**[1], **Sumaira Fida Abbasi**[1], **Mahmood Akhtar Kayani**[1]

**1** Cancer Genetics and Epigenetics Research Group, Department of Biosciences, COMSATS University Islamabad, Islamabad, Pakistan, **2** Department of Rheumatology, Military hospital, Rawalpindi, Pakistan

☯ These authors contributed equally to this work.

* ishrat.mahjabeen@comsats.edu.pk

**Editor:** Erika Di Zazzo, University of Molise Department of Medicine and Health Sciences "Vincenzo Tiberio": Universita degli Studi del Molise Dipartimento di Medicina e Scienze della Salute Vincenzo Tiberio, ITALY

**Data Availability Statement:** All relevant data are within the manuscript.

## Abstract

Mitochondrial sirtuins have diverse role specifically in aging, metabolism and cancer. In cancer, these sirtuins play dichotomous role as tumor suppressor and promoter. Previous studies have reported the involvement of sirtuins in different cancers. However, till now no study has been published with respect to mitochondrial sirtuins and glioma risks. Present study was purposed to figure out the expression level of mitochondrial sirtuins (*SIRT3, SIRT4, SIRT5*) and related genes (*GDH, OGG1-2α, SOD1, SOD2, HIF1α* and *PARP1*) in 153 glioma tissue samples and 200 brain tissue samples from epilepsy patients (taken as controls). To understand the role of selected situins in gliomagenesis, DNA damage was measured using the comet assay and oncometabolic role (oxidative stress level, ATP level and NAD level) was measured using the ELISA and quantitative PCR. Results analysis showed significant down-regulation of *SIRT4* (p = 0.0337), *SIRT5* (p<0.0001), *GDH* (p = 0.0305), *OGG1-2α* (p = 0.0001), *SOD1* (p<0.0001) and *SOD2* (p<0.0001) in glioma patients compared to controls. In case of *SIRT3* (p = 0.0322), *HIF1α* (p = 0.0385) and *PARP1* (p = 0.0203), significant up-regulation was observed. ROC curve analysis and cox regression analysis showed the good diagnostic and prognostic value of mitochondrial sirtuins in glioma patients. Oncometabolic rate assessment analysis showed significant increased ATP level (p<0.0001), NAD+ level [(NMNAT1 (p<0.0001), NMNAT3 (p<0.0001) and NAMPT (p<0.04)] and glutathione level (p<0.0001) in glioma patients compared to controls. Significant increased level of damage ((p<0.04) and decrease level of antioxidant enzymes include superoxide dismutase (SOD, p<0.0001), catalase (CAT, p<0.0001) and glutathione peroxidase (GPx, p<0.0001) was observed in patients compared to controls. Present study data suggest that variation in expression pattern of mitochondrial sirtuins and increased metabolic rate may have diagnostic and prognostic significance in glioma patients.

## Introduction

Gliomas are among the most common type of CNS tumors and epidemiological studies have stated that gliomas are 40% of all CNS and brain tumors [1–4]. On the basis of molecular and

**Funding:** The author(s) received no specific funding for this work.

**Competing interests:** The authors have declared that no competing interests exist.

histopathological features, gliomas are classified into low grade gliomas (LGG) and high-grade gliomas (HGG). Risk factors of gliomas are environmental factors, epigenetic and genetic factors. Genetic factors include the genetic variations/polymorphisms in metabolic/repairing pathway genes, apoptosis, and necroptosis [5, 6]. Most of the data published on genetic factors is present on mutations in nuclear region while the involvement of mitochondrial genes in glioma is lacking [5].

Mitochondria is very important organelle for cellular processes like energy production by oxidative phosphorylation, apoptosis, cell cycle proliferation etc. Reactive oxygen species (ROS) are highly produced in mitochondria and when this ROS levels are high, they can be harmful to cell, damages proteins, lipids and DNA hence results in mitochondrial dysfunction and carcinogenesis [7]. Brain and CNS consume a high percentage of energy and neurons are more susceptible to oxidative damage and ROS and has been found to be involved in neurodegenerative diseases [8]. Rate of mutation has been reported higher in mitochondrial DNA as it lacks the protective histone proteins and inefficient DNA repair mechanisms [9]. Furthermore, in glioma patients, another mitochondrial mechanism actively involved is necroptosis, which controlled the deep dyregulation of apoptosis signals. This dyregulation of programmed cell death ultimately results in gliomagenesis and tumor aggressiveness [6, 10].

Mitochondrial variations have also been found to link with carcinogenesis by deregulation of number of important mechanisms [11]. Among these pathways, variations of mitochondrial sirtuins genes have been suggested. Sirtuins, a family of orthologues of yeast silent information regulator 3 (*SIRT3*), 4 (*SIRT4*) and 5 (*SIRT5*) are important tumor suppressor genes located in mitochondria [12, 13]. Among these proteins, *SIRT3* is considered as guardian of mitochondria due to its role in deacetylation and metabolism [14]. Its deacetylation results in activation of proteins essential to control the oxidative stress and DNA damage/repair [15]. *SIRT4*, second mitochondrial sirtuin, has been reported to control the generation of reactive oxygen species (ROS) in mitochondria. Furthermore, it is actively involved in an NAD+ dependent protein ADP-ribosly transferase activity [16]. *SIRT5*, third mitochondrial sirtuin, has been reported as a mitochondrial NAD+ dependent deacetylase and involved in ROS detoxification and TCA cycle [17]. Previous studies have reported the involvement of mitochondrial sirtuins in regulation of important mitochondrial pathways such as ROS detoxification, repair, apoptosis/necroptosis and energy metabolism [18–20]. Mitochondrial sirtuins showed their involvement in above mentioned physiological processes in coordination with other genes. Kumar and Lombard (2015) has reported that molecular targets of mitochondrial sirtuin *SIRT3*, *SIRT4* and *SIRT5* have been identified in proteomic experiments [21]. These targets are such as *hypoxia inducible factor 1 subunit alpha (HIF1a)*, *glutamate dehydrogenase* (*GDH*), *Superoxide dismutase 1 (SOD1)*, *Superoxide dismutase 2 (*SOD2*), *mitochondrial 8-oxoguanine DNA glycosylase (OGG1-2α)* and *poly (ADP-ribose) polymerase 1 (PARP1)* [13]. Mitochondrial sirtuins regulate the mitochondria ROS level in coordination with *SOD1*, *SOD2*, *HIF1a* and *OGG1-2α* and any abnormality in this regulation will results initiation of carcinogenesis [13, 22–29]. Mitochondrial sirtuins also controls the other molecules by deacetylation such as it deacetylates the *GDH* and maintained its regulatory role in mitochondrial metabolism [28]. Previous studies have reported that mitochondrial sirtuins interact with the *PARP1* and control the regulation of DNA repairing, cell survival and cell death [30].

Several studies have reported the role of mitochondrial sirtuins and above-mentioned genes in different diseases [22–29]. However, no study has been yet published with respect to glioma. Thus, the current study is designed to investigate expression level of *SIRT3*, *SIRT4*, *SIRT5* and associated genes *SOD1*, *SOD2*, *OGG1-2α*, *PARP1*, *GDH* and *HIF1α* in glioma patients and to find out whether this expressional deregulation effects the risk of glioma. Metabolic assays and DNA damage will also be performed to assess the ATP, NAD+, glutathione and oxidative stress levels in glioma samples.

## Methodology

### Patients identification and sampling

Glioma patients were identified with the help of oncologists. Samples of different types and grades were collected by following the CNS WHO 2016 grading system. Study cohort included 153 glioma tissue samples from Pakistan Institute of Medical Sciences. In case of controls, 200 surgical section of brain tissue of epilepsy patients was collected from neurosurgical section of the above-mentioned hospital. Epilepsy patients are taken as controls as per guidelines of previously reported studied of brain tumoriogenesis [31, 32]. Demographic details of study cohort are given in Table 1. The surgically excised tissues of glioma patients and controls were recruited and stored at -80˚C. Histopathological reports for each sample was also obtained. Other factors like smoking, ethnicity, age, gender was recorded to link with disease.

### Ethical approval

The study was conducted with a prior approval from the institutional ethical review board of COMSTAS University (CUI) Islamabad. Members of this committee include Dean ORIC (Office of Research Innovation and Commercialization) Prof. Dr. Mahmood A Kayani (convener), Professor Dr Mustafa (Chairman Deptt of Biosciences), Dr. Faheem Tahir (Deputy Director, NIH) and Dr. Muhammad Qaiser Fatmi (Head of department). Informed written/ signed consent was obtained from participating individuals.

**RNA extraction and cDNA synthesis.** RNA was extracted from the fresh tissue samples by Trizol method as previously described by Afzal et al., [33]. Extracted RNA was visualized and quantified by the 1% agarose gel electrophoresis and spectrophotometer. RNA samples with 50 ng concentration were selected and proceeded for cDNA synthesis using the commercially available kit by Thermo Scientific (USA). Synthesized cDNA was confirmed by *β-actin* PCR and stored at -20˚C until further processing.

**Table 1. Demographic parameters of study cohort.**

| Parameters | Glioma patients N = 153 | Controls N = 200 |
|---|---|---|
| **Age** | | |
| <41 | 55 | 47 |
| >41 | 98 | 153 |
| **Gender** | | |
| Males | 69 | 98 |
| Females | 84 | 102 |
| **WHO grade** | | |
| I | 15 | |
| II | 69 | |
| III | 32 | |
| IV | 37 | |
| **Survival Status** | | |
| Under treatment/ cured | 102 | |
| Dead | 51 | |
| **Smoking Status** | | |
| Smokers | 86 | |
| Non- smokers | 67 | |
| **IR Exposure** | | |
| Yes | 59 | |
| No | 94 | |

## Expression analysis

Primers specific for the selected genes *SIRT3, SIRT4, SIRT5, SOD1, SOD2, OGG1-2α, PARP1, GDH, HIF1α* and *β-actin* was designed using the integrated DNA technology (IDT) as previously described by Ul Haq et al., [16]. Expression levels of above mentioned genes were measured using the Quantitative real time polymerase chain reaction (q-RT PCR). Step one Plus Thermal cycler (Applied Biosystems) was used for this analysis. Each reaction constituted of 4 μl SYBR green, 2 μl of primers, 1 μl cDNA and 3 μl of PCR water. q-RT PCR conditions included, 10 min of initial denaturation, 1 min of annealing for 40 cycles. Cyclic threshold (CT) values were taken for each sample and $2^{-\Delta\Delta Ct}$ method was used to calculate relative expression.

## Validation by NCBI GEO omnibus dataset

Involvement of these genes was assessed in genome wide expression in one dataset. The dataset containing glioma samples along with control samples was selected. The publicly available data to compare with our PCR results was obtained from NCBI Gene Expression Omnibus (GEO) database, (https://www.ncbi.nlm.nih.gov/geo/). Experiment type was Expression profiling by array with Accession number GSE:4290, Platform GPL570 (HG-U133_plus_2) Affymetrix Human Genome U133 Plus 2.0 Array [16].

## Comet assay

Single cell gel electrophoresis was carried out for the estimation of DNA damage in glioma patients and healthy controls. Procedure adopted was carried out from Akram et al., [34] with some modifications. Glioma tissue samples of different subtypes along with healthy control tissues were used. The tissues were homogenized in saline solution and single cell suspension was prepared which was proceeded for detection of overall DNA damage in the tissues. Glass slides used in the procedure were labelled according to different glioma types. Three layered procedure was used the samples were sandwiched in the layers of low melting point (LMP) agarose gel. Whole area under coverslip was visualized and scored by fluorescent microscope. After running, mean %DNA damage, mean OTM (olive tail moment) and mean tail length were obtained and further processed for analysis. Comets were analyzed in Metafer 4 (MetaSystems).

## Metabolic assays

**ATP level determination.** Glioma tissue sample and controls taken from patients were treated with saline solution to prepare tissue lysate. Three aliquots were prepared, one aliquot was used for the measurement of ATP level determination, as described by Afzal et al., [33]. Second aliquot was used for oxidative stress measurement and third for DNA damage assessment. For ATP level measurement, cell lysate of glioma and healthy controls were dispensed in micro titer plate and Cell titer glo (CTG) buffer was then added in the micro titer plate and Luminescence was recorded and analyzed.

**Oxidative stress determination.** Glutathione level was measured from isolated protein lysate using Glutathione Assay Kit II (Calbiochem) according to manufacturer protocol provided with kit, as described by Afzal et al., [33].

Oxidative stress was measured by the cell lysate of the glioma samples and healthy controls. Oxidative stress was evaluated by quantitative analysis of three antioxidant enzymes including superoxide dismutase (SOD), catalase (CAT), and glutathione peroxidase (GPx). Enzyme analysis was carried out using human specific ELISA kits purchased from Abcam (UK).

Absorbency of enzymes was checked in microplate reader (Platos R496, AMP Diagnostics), and sample concentrations were read from calibration curve.

**NAD level determination.** NAD level was also measured by designing the primers for three important enzymes regulating NAD levels, they are NAMPT, NMNAT1 and NMNAT3. The q-RT-PCR procedure and program designing was same as stated above for expression analysis.

**Data analysis.** Statistical tools from SPSS and GraphPad Prism were used to analyze the data. *SIRT3*, *SIRT4*, *SIRT5*, *SOD1*, *SOD2*, *GDH*, *mt OGG1-2α*, *PARP1* and *HIF1a* expression values were compared using student t-test, DNA damage values were compared using student t-test and one sample t-test while NAD+ and ATP values were also compared using student t-test. Gene to gene and gene to different histopathological parameters were compared by applying spearman correlation and diagnostic value was checked by ROC analysis. Survival analysis was performed by Kaplan Maier analysis.

## Results

### Expressional analysis

Relative expression of these selected genes was analyzed in 153 glioma samples along with 200 controls using qPCR. Significant down-regulated expression of *SIRT4* (p = 0.033), *SIRT5* (p<0.0001), *GDH* (p = 0.03), *OGG1-2α* (p<0.0001), *SOD1* (p<0.0001) and *SOD2* (p<0.0001) was observed in glioma patients compared to controls as shown in Fig 1A and 1C. In case of *SIRT3* (p = 0.03), *HIF1α* (p = 0.03) and *PARP1* (p = 0.02) gene, significant upregulated expression was observed in glioma patients compared to control samples as shown in Fig 1A and 1C.

Demographic details of selected glioma patients and controls are given in Table 1. In case of grading, significant upregulation of *SIRT3* (p = 0.0128) was observed in high grade glioma (HGG) compared to low grade glioma (LGG) as shown in Fig 1A. All other genes such as *SIRT4* (p = 0.0129; Fig 2B), *SIRT5* (p = 0.0461; Fig 2C), *SOD1* (p = 0.0239; Fig 2D), *SOD2* (p = 0.0066; Fig 2E), *OGG1-2α* (p = 0.0096, Fig 2F), *HIF1α* (p = 0.0149; Fig 2G), *GDH* (p = 0.0469; Fig 2H) and *PARP1* (p = 0.0087; Fig 2I) were significantly downregulated in HGGs as compared to LGG. In case of smoking status, *SIRT3* was found significantly upregulated (p = 0.0097; Fig 1A) and *SIRT4* (p = 0.0015; Fig 2B), *OGG1-2α* (p = 0.0026; Fig 2F), *SOD1* (p = 0.0445; Fig 2D) and *SOD2* (p = 0.0267; Fig 2E) were significantly downregulated in smokers compared to non-smokers. While non-significant results were observed in case of *SIRT5* (p = 0.3604; Fig 2C), *GDH* (p = 0.204; Fig 2H), *HIF1α* (p = 0.1500; Fig 2G) and *PARP1* (p = 0.4626; Fig 2I) in smokers compared to nonsmokers.

**Validation by NCBI GEO Omnibus dataset.** Relative expression of our selected genes was validated from dataset GSE4290, titled: Expression data of glioma from Henry Ford hospital, at first data was normalized and analyzed in GraphPad Prism. Data set consisted of 157 glioma samples and 23 Epilepsy samples which were used as controls. 157 tumor samples include 26 astrocytoma, 50 oligodendrogliomas and 81 glioblastomas. Data analysis showed that *SIRT4* (p<0.0001), *SIRT5* (p = 0.0076), *OGG1-2α* (p<0.0001), *SOD1* (p<0.0001) and *SOD2* (p<0.0001) showed significant downregulation as shown in Fig 1B and 1D. while *GDH* (p = 0.6776) showed non significantly downregulated results (Fig 1D). In case of *SIRT3* (p = 0.0142) and *HIF1α* (p = 0.0385) significant upregulation was observed (Fig 1B and 1D) while non-significant upregulated expression was observed in *PARP1* (p = 0.203) in glioma samples as compared to healthy samples. Results are shown in Fig 1B and 1D.

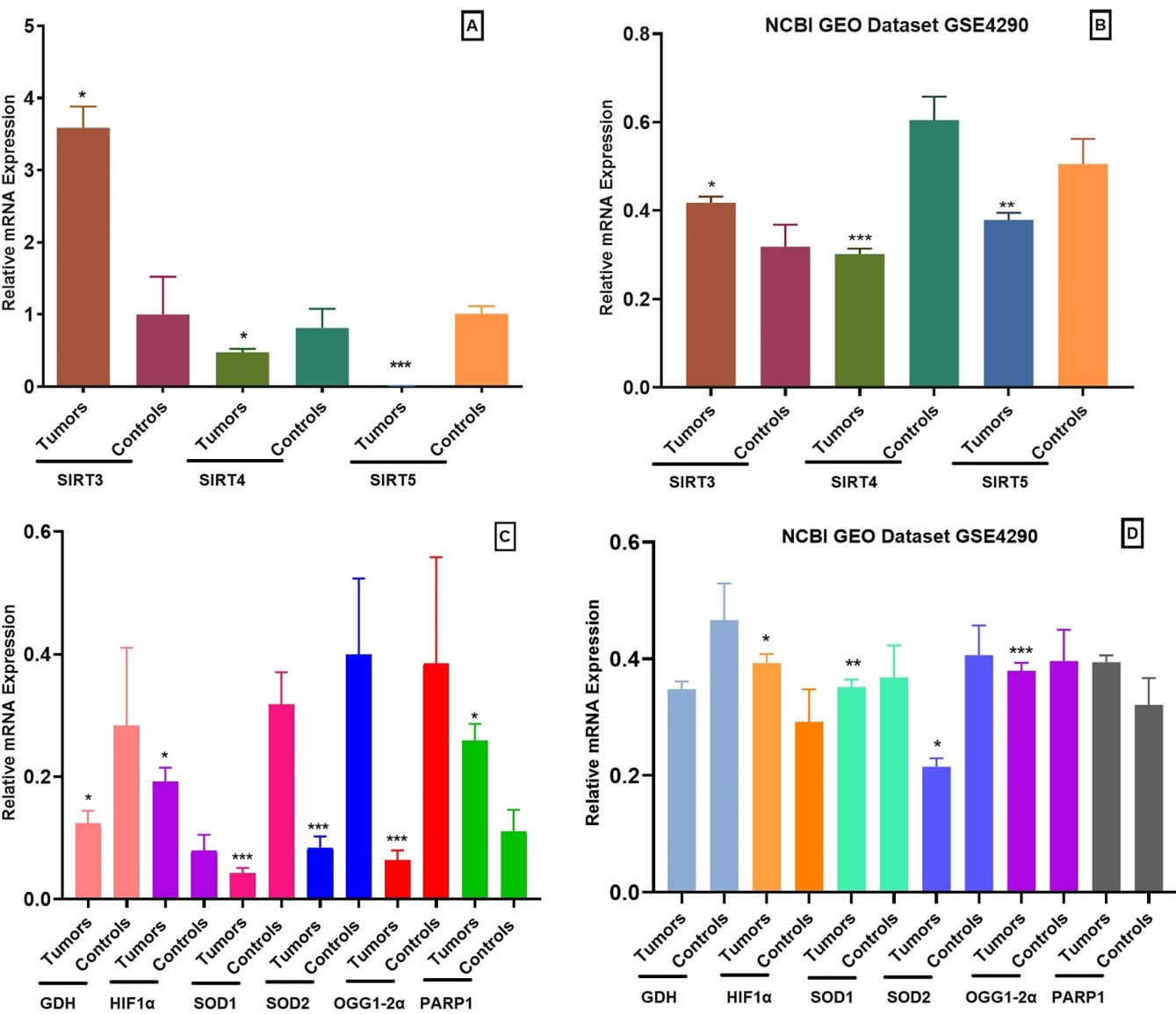

**Fig 1. q-PCR analysis of mitochondrial sirtuins and relative genes in glioma patients compared to controls.** (A) Relative expression levels of mitochondrial sirtuins *SIRT3*, *SIRT4* and *SIRT5* in glioma patients compared to controls. (B) Relative expression levels of mitochondrial sirtuins *SIRT3*, *SIRT4* and *SIRT5* in publicly available dataset of glioma tumors compared to controls. (C) Relative expression levels of mitochondrial sirtuins related genes such as *GDH*, *HIF1a*, *SOD1*, *SOD2*, *OGG1-2α* and *PARP1* in glioma patients compared to controls. (D) Relative expression levels of *GDH*, *HIF1a*, *SOD1*, *SOD2*, *OGG1-2a* and *PARP1* in publicly available dataset of glioma tumors compared to controls. P<0.05*, p<0.01**, p<0.001***.

## Comet assay

For DNA damage assessment, comet assay was performed for glioma samples and uninvolved healthy controls. In this assay number of comets were assessed in glioma sections Fig 3A2 compared to the healthy controls Fig 3A1. Data analysis showed significantly higher number of comets in tumor samples (p = 0.04) compared to the controls (Fig 3B).

In the assay different parameters were also calculated which are %DNA in tail and it showed significant higher %DNA in tail in glioma patients compared to controls (p = 0.0005), HGG compared to LGG (p = 0.0374) and smokers in comparison with non-smokers (p = 0.0320) as shown in Fig 3C and 3F. In case of olive tail moment, it was observed significantly higher in

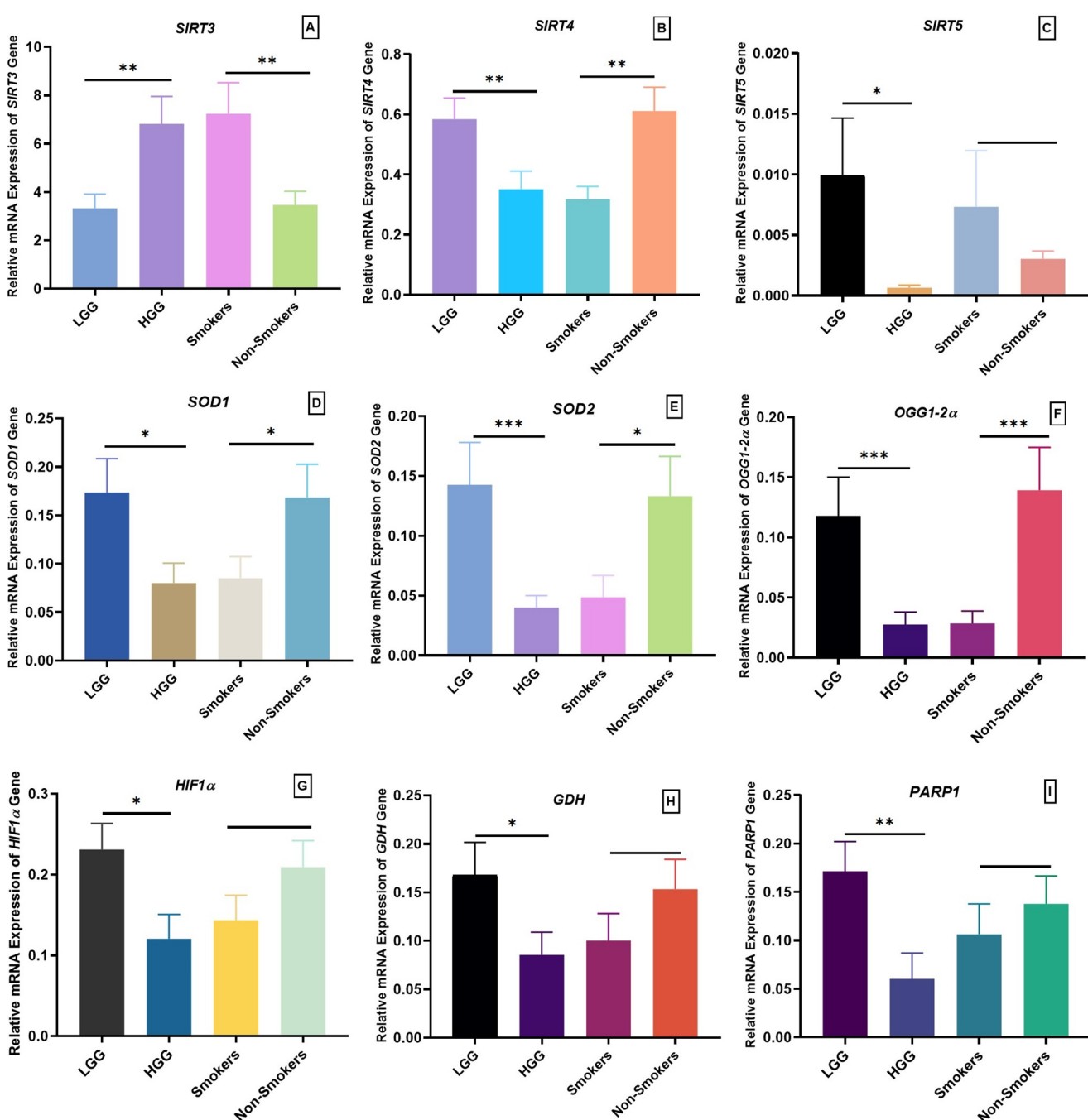

**Fig 2. Association of relative expression of mitochondrial sirtuins and relative genes with histopathological parameters of glioma patients.** (A) Relative expression of *SIRT3* gene in low grade glioma (LGG) vs high grade glioma (HGG) and in smokers vs nonsmokers. (B) Relative expression of *SIRT4* gene in low grade glioma (LGG) vs high grade glioma (HGG) and in smokers vs nonsmokers. (C) Relative expression of *SIRT5* gene in low grade glioma (LGG) vs high grade glioma (HGG) and in smokers vs nonsmokers. (D) Relative expression of *SOD1* gene in low grade glioma (LGG) vs high grade glioma (HGG) and in smokers vs nonsmokers. (E) Relative expression of *SOD2* gene in low grade glioma (LGG) vs high grade glioma (HGG) and in smokers vs nonsmokers. (F) Relative expression of *OGG1-2a* gene in low grade glioma (LGG) vs high grade glioma (HGG) and in smokers vs nonsmokers. (G) Relative expression of HIF1a gene in low grade glioma (LGG) vs high grade glioma (HGG) and in smokers vs nonsmokers. (H) Relative expression of *GDH* gene in low grade glioma (LGG) vs high grade glioma (HGG) and in smokers vs nonsmokers. (I) Relative expression of *PARP1* gene in low grade glioma (LGG) vs high grade glioma (HGG) and in smokers vs nonsmokers. $P < 0.05^*$, $p < 0.01^{**}$, $p < 0.001^{***}$.

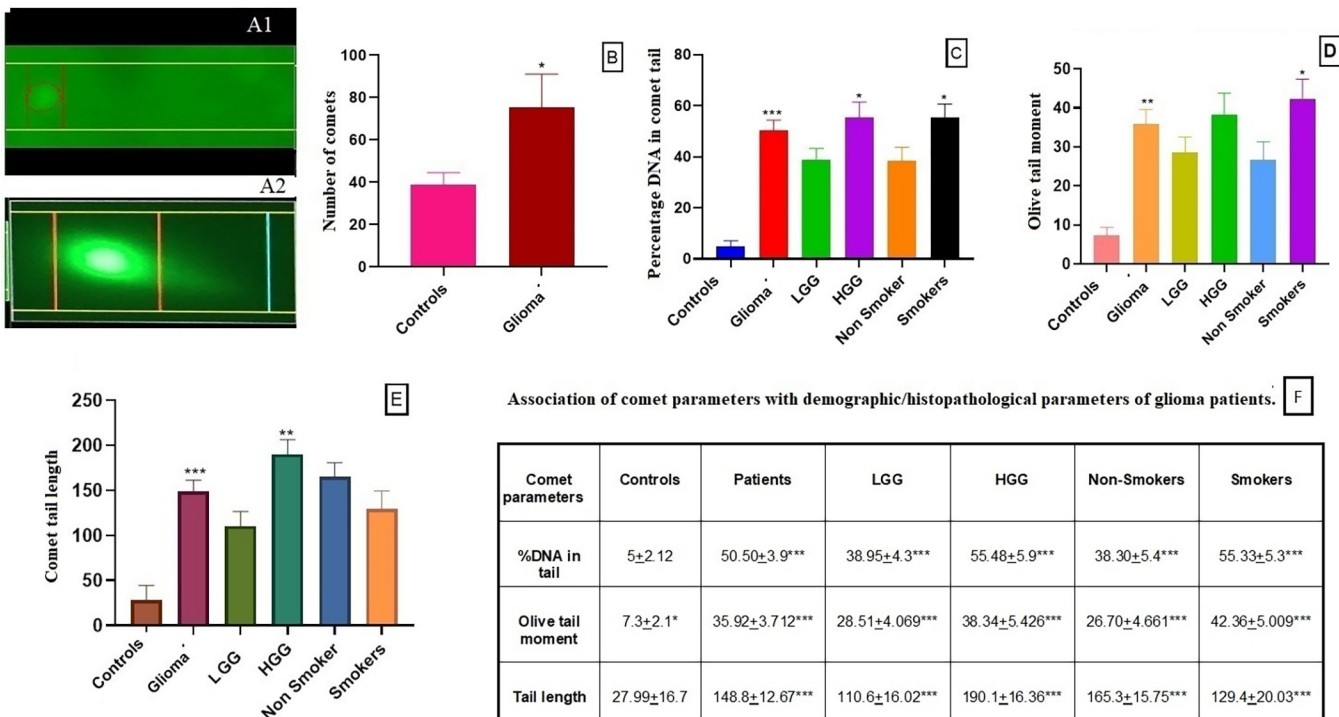

**Fig 3. Comet analysis of glioma patients and controls.** (A1) Comet tail and length in controls, (A2) Comet tail and length in glioma, (B) number of comets in glioma patients vs controls, (C) percentage DNA in comet tail in glioma patients vs controls, with respect to different grades and smoking status of glioma patients, (D) olive tail moment in glioma patients vs controls, with respect to different grades and smoking status of glioma patients. (E) comet tail length in glioma patients vs controls, with respect to different grades and smoking status of glioma patients, (F) association of comets with different demographic/histopathological parameters of glioma patients.

glioma tissue samples compared to controls (p = 0.008), smokers compared to non-smokers (p = 0.0266) and non-significantly higher in case of HGG as compared to LGG (p = 0.1582) as shown in Fig 3D and 3F. Tail length was also assessed in patients and controls, tail length was observed higher in glioma patients compared to controls (p = 0.0005), HGG compared to LGG (p = 0.0011) while non-significantly less in smokers as compared to non-smokers (p = 0.1605) as shown in Fig 3E and 3F, Table 2.

## Metabolic assays

ATP level was determined in glioma samples compared to the healthy controls. Analysis showed that ATP level was found significantly higher (p<0.03) in glioma compared to the control samples as shown in Fig 4A. ATP levels were also found to be significantly higher in HGG as compared to LGG and non-significantly low in smokers as compared to non-smokers, as shown in Fig 4B.

Glutathione level was also assessed, and it was found to be significantly higher in glioma patients compared to controls (p<0.0001) and in HGG as compared to LGG (p<0.001), as shown in Fig 4C and 4D. While in case of smokers, glutathione level was found to be non-significantly higher as compared to non-smokers Fig 4D.

NAD+ level was also determined by measuring the relative expression levels of NMNAT1, NMNAT3 and NAMPT. NMNAT1 was found significantly upregulated in gliomas (p<0.001) compared to controls and in HGG (p<0.001) compared to LGG patients. While in case of smokers, NMNAT1 was found non significantly upregulated compared to non-smokers as

**Table 2. Multivariant COX regression analysis of selected microRNAs in glioma patients.**

| Parameters | B | SE | Wald | DF | Sig | Exp (B) |
|---|---|---|---|---|---|---|
| *SIRT3* | 1.724 | 0.524 | 10.827 | 1 | 0.0001 | 5.61 |
| *SIRT4* | 1.451 | 0.441 | 10.827 | 1 | 0.001 | 4.27 |
| *SIRT5* | 1.297 | 0.462 | 7.879 | 1 | 0.005 | 3.66 |
| *SOD1* | 0.190 | 0.112 | 2.874 | 1 | 0.09 | 1.21 |
| *SOD2* | 0.418 | 0.231 | 3.283 | 1 | 0.07 | 1.52 |
| *GDH* | 1.095 | 0.425 | 6.634 | 1 | 0.01 | 2.99 |
| *OGG1-2a* | 1.187 | 0.461 | 6.634 | 1 | 0.003 | 3.28 |
| *PARP1* | 1.515 | 0.460 | 10.827 | 1 | 0.001 | 4.55 |
| *HIF1α* | 1.795 | 0.461 | 15.136 | 1 | 0.0001 | 6.02 |
| Age | 0.212 | 0.310 | 0.467 | 1 | 0.495 | 1.236 |
| Gender | -0.562 | 0.319 | 3.09 | 1 | 0.078 | 0.570 |
| Smoking | -0.587 | 0.315 | 3.461 | 1 | 0.063 | 0.556 |
| Grade | -0.628 | 0.318 | 3.908 | 1 | 0.04 | 0.534 |
| IR exposure | -0.514 | 0.303 | 2.880 | 1 | 0.09 | 0.598 |

SE = standard error; DF = degree of freedom; Sig = significance $p < 0.05$

shown in Fig 5A. Second selected gene, NMNAT3 was found significantly upregulated in gliomas ($p < 0.001$) compared to controls and in HGG ($p < 0.001$) compared to LGG as shown in Fig 5B. while in case of smokers, NMNAT3 was found non significantly upregulated compared to non-smokers Fig 5B. Third selected gene, NAMPT was found significantly upregulated in gliomas compared to controls ($p < 0.04$) and in HGG as compared to LGG ($p < 0.001$) as shown in Fig 5C. while in case of smokers NAMPT was found non significantly upregulated as compared to non-smokers as shown in Fig 5C.

## Measurement of oxidative stress

Oxidative stress was evaluated in study cohort by measuring the three main antioxidant enzymes which includes superoxide dismutase (SOD), catalase (CAT) and glutathione peroxidase (GPx) as shown in Fig 6. SOD ($p < 0.001$), CAT ($p < 0.001$) and GPx ($p < 0.001$) levels were found significantly down regulated in glioma patients compared to controls, as shown in Fig 6A. Further analysis showed the significant downregulated expression of SOD ($p < 0.001$), CAT ($p < 0.001$) and GPx ($p < 0.001$) in HGG compared to LGG as shown in Fig 6B and in smokers [SOD ($p < 0.001$), CAT ($p < 0.02$) and GPx ($p < 0.04$)] compared to non-smokers (Fig 6C).

## Spearman correlation

Gene to gene correlation was performed by spearman mentioned in Table 3. Negative correlation was observed between *SIRT3* and *SIRT4* (r = -0.2213 p = 0.050), *SIRT3* and *SIRT5* (r = -0.04447 p = 0.697), *SIRT3* and *SOD2* (r = -0.03706 p = 0.746), *SIRT3* and *HIF1α* (r = -0.09681 p = 0.396), *SIRT4* and *SOD1* (r = -0.03940 p = 0.712), *SIRT4* and *GDH* (r = -0.1180 p = 0.268), *SIRT4* and *SOD1* (r = -0.03940 p = 0.712), *SIRT4* and *HIF1α* (r = -0.09176 p = 0.390), *SIRT5* and *SOD1* (r = -0.0529 p = 0.620), *SIRT5* and *HIF1α* (r = -0.01410 p = 0.89),*SIRT5* and *PARP1* (r = -0.0927 p = 0.89),*SIRT5* and *OGG1-2α* (r = -0.1080 p = 0.311), *SIRT5* and *SOD2* (r = -0.0456 p = 0.670), *SIRT5* and *GDH* (r = -0.01657 p = 0.877), *SOD1* and *GDH* (r = -0.2783, p = 0.015*), *GDH* and *HIF1a* (r = -0.04, p = 0.712), *GDH* and *PARP1* (r = -0.0606, p = 0.603),

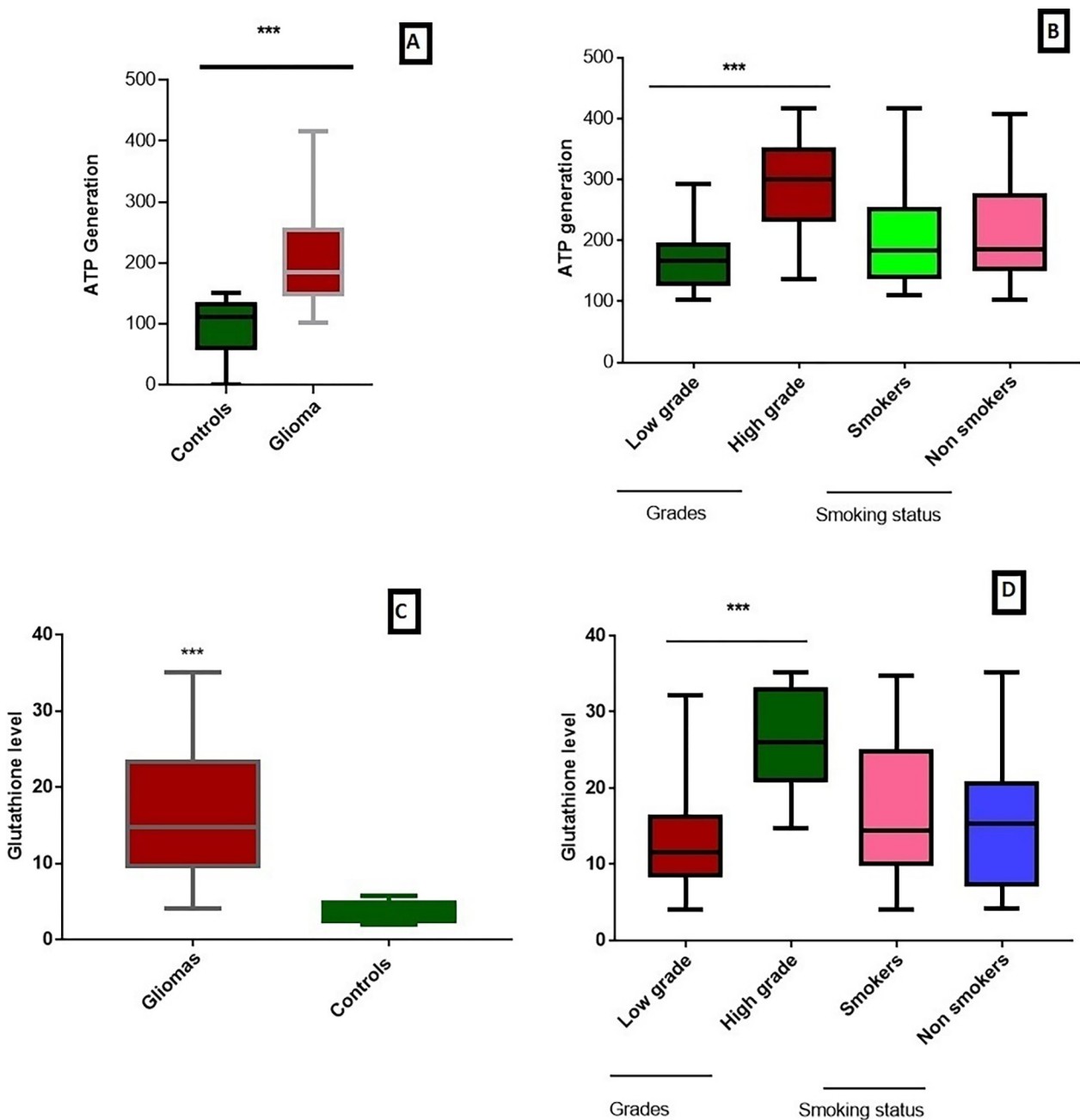

**Fig 4. Measurement of ATP and glutathione levels in glioma vs controls.** (A) ATP level in glioma vs controls. (B) ATP level in in low grade glioma (LGG) vs high grade glioma (HGG) and in smokers vs nonsmokers. (C) Glutathione level in glioma vs controls. (D) Glutathione level in in low grade glioma (LGG) vs high grade glioma (HGG) and in smokers vs nonsmokers. P<0.05*, p<0.01**, p<0.001***.

*SOD1* and *HIF1a* (r = -0.079, p = 0.486), *SOD2* and *HIF1a* (r = -0.0149, p = 0.896), *SOD1* and *SOD2* (r = -0.0149, p = 0.483) in glioma patients.

Furthermore, positive association was observed between *SIRT3* and *SOD1* (r = 0.06661 p = 0.560), *SIRT3* and *PARP1* (r = 0.2825 p = 0.012*), *SIRT3* and *GDH* (r = 0.0067 p = 0.953), *SIRT3* and *OGG1-2α* (r = 0.2401 = 0.033*), *SIRT4* and *SIRT5* (r = 0.04925 p = 0.647), *SIRT4* and *PARP1* (r = 0.01022 p = 0.924), *SIRT4* and *SOD2* (r = 0.02412 p = 0.821), *SOD1* and

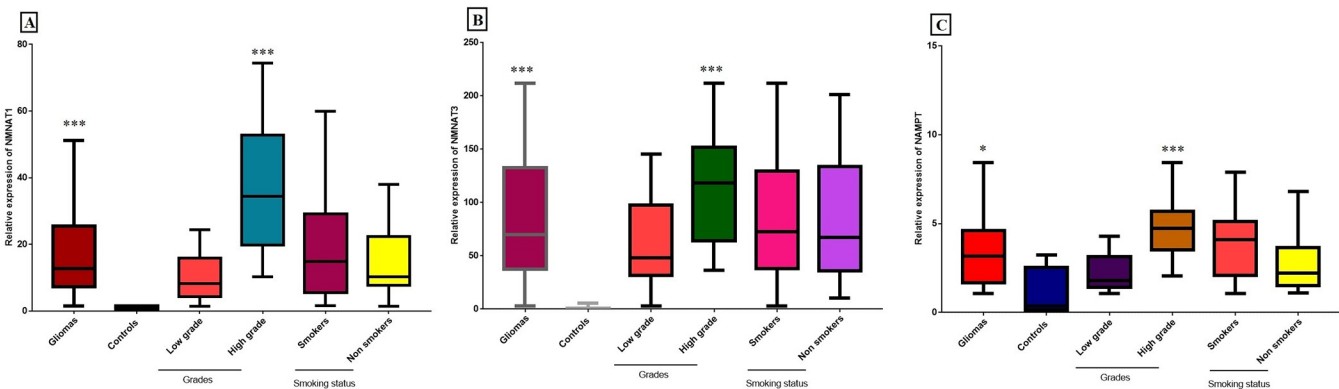

**Fig 5. NAD level was measured by relative expression of NAD enzymes.** (A) Relative expression level NMNAT1 gene in glioma and controls. Association of relative expression level of NMNAT1 with different grades and smoking status of glioma patients. (B) Relative expression level NMNAT2 gene in glioma and controls. Association of relative expression level of NMNAT2 with different grades and smoking status of glioma patients. (C) Relative expression level NAMPT gene in glioma and controls. Association of relative expression level of NAMPT with different grades and smoking status of glioma patients. P<0.05*, p<0.001***.

*PARP1* (r = 0.0147, p = 0.129), *GDH* and *OGG1-2α* (r = 0.066, p = 0.568), *SOD2* and *OGG1-2a* (r = 0.3745, p = 0.001***), *OGG1-2α* and *PARP1* (r = 0.009846, p = 0.932), *OGG1-2α* and *HIF1a* (r = 0.01133, p = 0.922), *PARP1* and *HIF1a* (r = 0.0147, p = 0.898), *GDH* and *SOD2* (r = 0.043, p = 0.711), *SOD1* and *OGG1-2α* (r = 0.044, p = 0.699), *SOD2* and *PARP1* (r = -0.2577*, p = 0.02) in glioma patients.

## ROC analysis

To assess the diagnostic value of these selected genes, ROC curve analysis was performed as shown in Fig 7. ROC curve showed the area under the curve (AUC) and 95% confidence interval (CI). Analysis showed that calculated area under the curve was 82% for *SIRT3* (AUC = 0.8203; 95%CI = 0.672–0.9685; p<0.0322, Fig 7A), 53% for *SIRT4* (AUC = 0.533; 95% CI = 0.336–0.7307, p<0.0337, Fig 7B), 100% for *SIRT5* (AUC = 1.000; 95%CI = 1.000–1.000; p<0.0001, Fig 7C), 61% for *GDH* (AUC = 0.6127, 95%CI = 0.4720–0.7533, p<0.1582 Fig 7D), 69% for *HIF1a* (AUC = 0.6979; 95%CI = 0.5557–0.8401; p value<0.0155 Fig 7E), 94% for *SOD1*(AUC = 0.9469; 95%CI = 0.9034–0.9903, p<0.0001 Fig 7F), 87% for *SOD2*

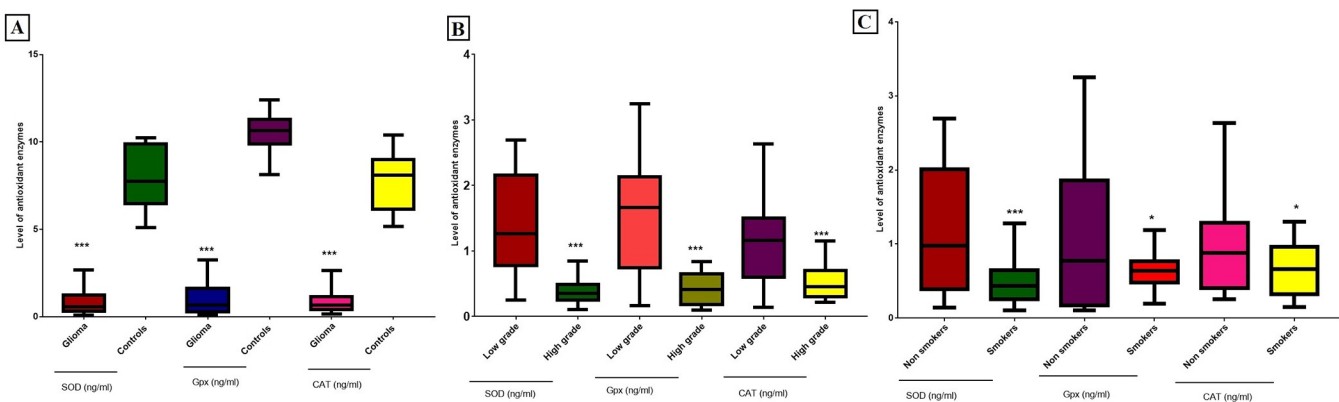

**Fig 6.** Measurement of oxidative stress by determining the level of antioxidant enzyme SOD, Gpx and CAT in (A) glioma patients vs controls, (B) patients with low grade vs high grade tumor, (C) patients with smoking status and non-smoking status. P<0.05*, p<0.001***.

**Table 3. Gene-gene correlation of selected genes in glioma patients.**

| Genes | SIRT4 | SIRT5 | SOD1 | SOD2 | HIF1α | PARP1 | GDH | OGG1-2α |
|---|---|---|---|---|---|---|---|---|
| SIRT3 | -0.2213* | -0.0444 ns | 0.06661 ns | -0.03706 ns | -0.09681 ns | 0.2825* | 0.0067 ns | 0.2401* |
| SIRT4 | | 0.0492 ns | -0.03940 ns | 0.0241 ns | -0.09176 ns | 0.01022 ns | -0.1180 ns | -0.05450 ns |
| SIRT5 | | | -0.05295 ns | -0.04560 ns | -0.01410 ns | -0.09277 ns | -0.01657 ns | -0.1080 ns |
| SOD1 | | | | -0.1118 ns | -0.417*** | -0.212** | 0.23** | -0.098 ns |
| SOD2 | | | | | 0.39*** | 0.0714 ns | 0.02 ns | 0.1323 ns |
| HIF1α | | | | | | 0.2698*** | 0.0397 ns | 0.08 ns |
| PARP1 | | | | | | | -0.08 ns | 0.1356 ns |
| GDH | | | | | | | | -0.06 ns |

Spearman correlation was performed. ns = non-significant (p>0.05).

*p<0.05

**p<0.01

***p<0.001

(AUC = 0.8749, 95%CI = 0.8014–0.9484, p<0.0001, Fig 7G), 69% for *OGG1-2α* (AUC = 0.6973, 95%CI = 0.5539–0.8407, p<0.0234, Fig 7H) and 67% for *PARP1* (AUC = 0.6733, 95%CI = 0.5469–0.7996, p<0.0294 Fig 7I) in Glioma patients.

## Survival analysis

Kaplan-Meier analysis was performed to check the prognostic value of mitochondrial sirtuins and their associated genes. Survival analysis showed that deregulation of SIRT3 (p<0.0081; Fig 8A), *SIRT4* (p<0.0381; Fig 8B), *SIRT5* (p<0.0151; Fig 8C), *GDH* (p<0.0108; Fig 8F), *OGG1-2α* (p<0.05; Fig 8G), *PARP1* (p<0.0270; Fig 8H) and *HIF1α* (p<0.0053; Fig 8I) was found associated with significant decrease survival of glioma patients as shown in Fig 8. However, nonsignificant difference was observed in case of SOD1 (Fig 8D) and SOD2 (Fig 8E).

## Discussion

Gliomas are most common intra-axial brain tumors originated from glial cells [5]. Risk factors for glioma include ionizing radiations, diet, exposure to pesticides, viruses, and genetic factors [35]. Genetic susceptibility also plays an important role in carcinogenesis and different pathways work to maintain genetic mechanisms. These pathways include mitochondrial DNA encoded genes as well as nuclear-encoded mitochondrial proteins [36]. Sirtuin family is among nuclear-encoded mitochondrial proteins, they are family of orthologues of yeast silent information regulator 3 (*SIRT3)*, 4 (*SIRT4)* and 5 (*SIRT5)*. Many studies have been proposed to discuss the role of sirtuins in cancers but still no study has been published to show their role with gliomas. The aim of present study was to elucidate the role of expression deregulations of mitochondrial sirtuin genes and co-expression with their associated proteins (*SOD1*, *SOD2*, *OGG1-2α*, *PARP1*, *GDH and HIF1α*) in glioma. Diagnostic/prognostic value of these genes was also analyzed in the glioma samples to check the role of these genes in glioma diagnosis. Metabolic assay and oxidative stress assay were also performed to assess the nonmetabolic role of selected genes in glioma tumoriogenesis.

*SIRT3* is NAD+ dependent mitochondrial deacetylase, it is usually present in nucleus but in stress condition it is transported to mitochondria where it gets trimmed and activates. It regulates many important biological functions like metabolic control, gene expression, aging and many diseases like cancer [37]. Many studies have reported that metabolic programs are linked to tumorigenesis, as *SIRT3* is an important mediator of mitochondrial metabolic pathways so

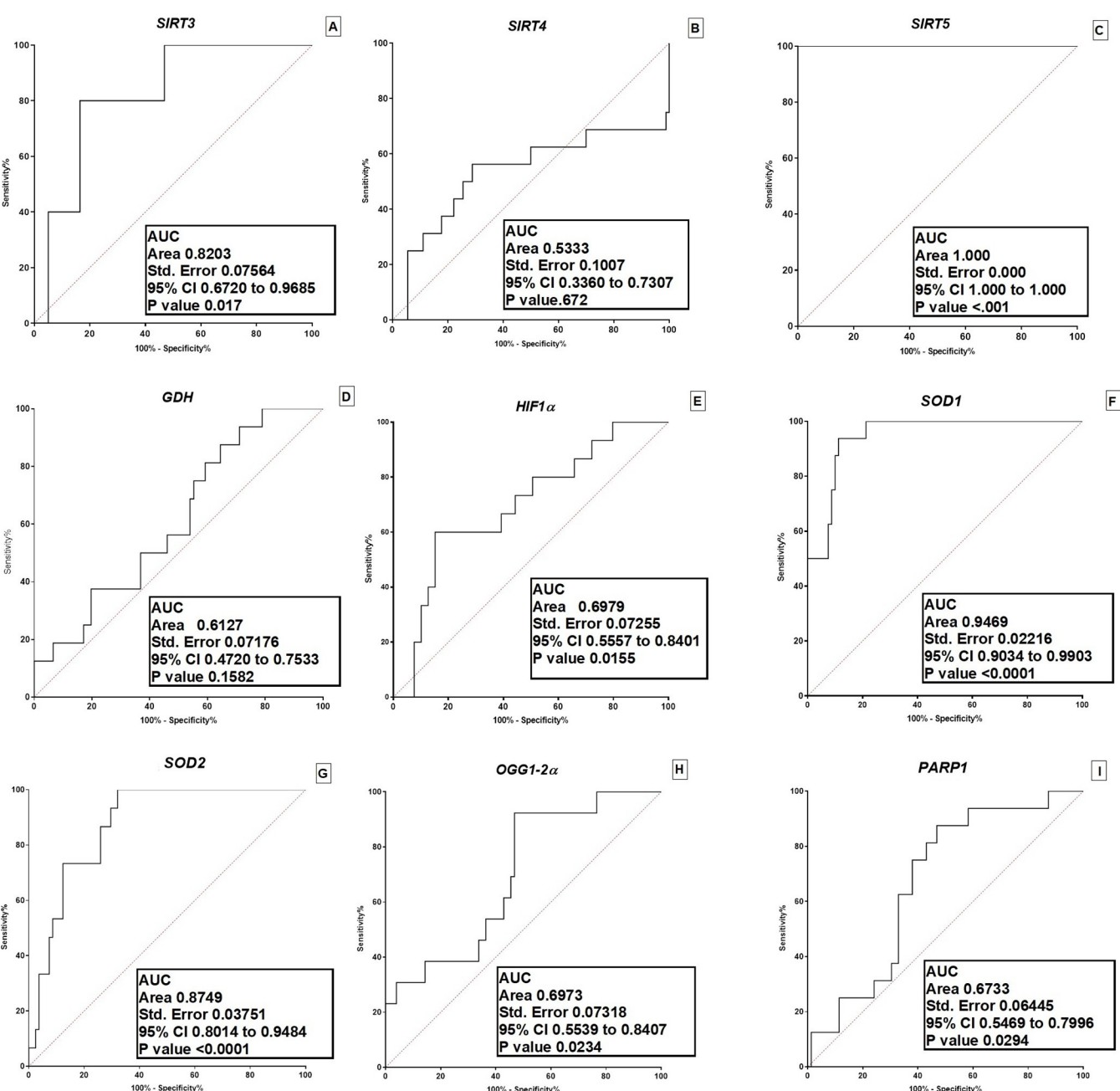

**Fig 7. Assessment of diagnostic of selected mitochondrial sirtuins and related genes in glioma patients.** (A) Area under curve of SIRT3, (B) area under curve of SIRT4, (C) area under curve of SIRT5, (D) area under curve of GDH, (E) area under curve of HIF1a, (F) area under curve of SOD1, (G) area under curve of SOD1, (H) area under curve of SOD2, (I) area under curve of OGG1-2a and (J) area under curve of PARP1 in glioma patients.

its overexpression and down-regulation is also linked to the tumor formation or tumor suppression [11]. In present study significant upregulation of *SIRT3* was observed and this upregulation was found linked to the tumor aggressiveness and poor survival. Overexpression of *SIRT3* has been reported in many cancers like in ovarian cancer [38], colon cancer [39] and colorectal cancer [40]. Reason behind the upregulation of SIRT3 gene and increased carcinogenesis needs to be explored. However, our previous study on the involvement of SIRT3 gene

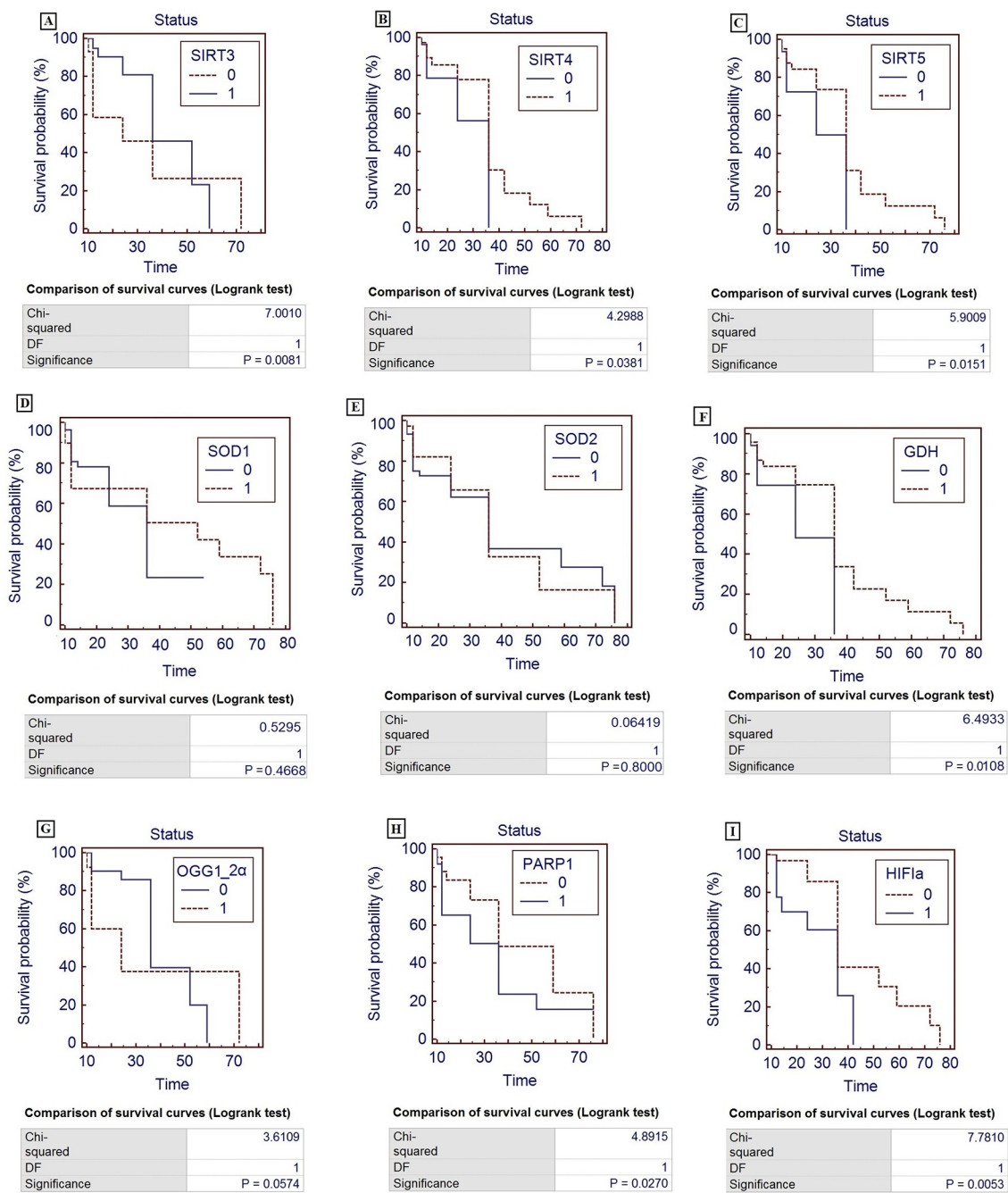

**Fig 8. Kaplan meier analysis of selected mitochondrial sirtuins and related genes in glioma patients.** (A) Survival curve of SIRT3, (B) survival curve of SIRT4, (C) survival curve of GDH, (D) survival curve of SOD1, (E) survival curve of SOD2, (F) survival curve of GDH, (G) survival curve of OGG1, (H) survival curve of PARP1, (I) survival curve of HIF1a. 0 represents downregulation, 1 represents upregulation. DF is degree of freedom, level of significance p<0.05.

in carcinogenesis showed that genetic polymorphism in said gene might results in mRNA splicing, epigenetics variations and impaired activity [41]. Previous studies have reported that upregulation of SIRT3 gene may results in reduction in apoptotic power and increased survival capacity of tumor cells, which ultimately leads to increased aggressiveness of tumor [42, 43].

*SIRT4 i*s also found working in mitochondria, it controls the working of glutamate dehydrogenase (GDH), which ultimately controls the glutamate and glutamine activity by ADP-ribosylation. *SIRT4* is found working as suppressor of glutamine under stressful condition, like oxidative stress and stabilizes the genomic stability [44]. Results from present study showed the significant downregulation of *SIRT4* in glioma patients compared to healthy controls depicting its role as a tumor suppressor. Previous studies have also showed the downregulation of *SIRT4* in gastric [45], thyroid [46] and breast cancer [47]. Mahjabeen & Kayani (2017) have reported that reason behind the downregulation of SIRT4 gene can be the variation in coding/noncoding region of said gene, loss of functionality, abnormal metabolic rate, and increased carcinogenesis [17].

*SIRT5* is another member of mitochondrial sirtuins, it is also found working in deacetylase activity. *SIRT5* binds to glutaminase, which is first enzyme for glutamine metabolism [48]. In this study SIRT5 is found to be downregulated in glioma patients as compared to the healthy controls depicting its role as a tumor suppressor. In previous studies *SIRT5* has been found downregulated in different cancers like liver cancer [48] gastric cancer [49] and hepatocellular carcinoma [50]. Previous studies have reported the upregulation of SIRT5 gene in breast [51], ovarian [20] and liver cancer [48]. Reason behind the downregulated and upregulated expression of SIRT5 gene is not clear yet. However, previous study has reported that high SNPs frequency in coding/noncoding region of SIRT5 gene resulted in deregulation of said genes, which ultimately results in increased reactive oxygen species, oxidative stress and carcinogenesis [52, 53].

SOD1 is mitochondrial protein and act as an antioxidant enzyme. It is involved in formation of hydrogen peroxide to superoxide. SOD1 is a cytosolic protein but is also localized in inter-membrane spaces of mitochondria [54, 55]. In present study SOD1 expression was found downregulated in gliomas compared to healthy control tissues which are in line with the previously reported findings. In previous studies, SOD1 has found to be overexpressed in numerous cancers like NSCLC [22], leukemia [56] and breast cancer [57], the overexpression is linked to poor survival of cancer cell [22]. Although *SOD1* is highly expressed and act as oncogene in many cancers but in pancreatic cancer cells *SOD1* has differentially expressed. It indicated that it is tissue-specific and play different and opposite roles in different types of cancers. High *SOD1* expression promotes tumor cell proliferation, it can be, by reduced antioxidant protein activity leading to increased intracellular H2O2. *SOD1* inhibition increases reactive oxygen species levels and results in increased DNA double strand breaks and carcinogenesis [22].

SOD2 is a key antioxidant, and it is localized in mitochondria and is found to act as potential tumor suppressor gene in carcinogenesis [41]. SOD2 has also showed differential expression in different cancers such as cervical [58], lung [59], brain [60], colon [61], breast [62], gastric [63] and salivary cancer [64]. This overexpression has been found linked to metastasis, poor prognosis and lower survival. In another study SOD2 expression has been found to be downregulated in cancerous cells as compared to control cells [24]. In present study SOD2 has found to be downregulated, one of previous study has demonstrating the similar findings as it has been reported that SOD2 showed downregulated expression in breast, esophageal and pancreatic cancers [65].

*OGG1-2α* is DNA repair gene and low expression of DNA repair gene results in reduced DNA repair activity resulting in more oxidative imbalance [66]. In previous studies *OGG1-2α* has been found to be down regulated in carcinogenic cells compared to controls suggesting the systems of excision of bases operate without any disability [67]. It has been found downregulated in colorectal cancer [26] and gastric cancer, which might be due to genetic heterogeneity in *OGG1-2α* gene because of environmental, genetic, and epigenetic factors [68].

Overexpression of *OGG1-2α* has been found in one study in lung cancer which is because of the polymorphism Ser326Cys. It has been found that variation of Cys326/Cys326 genotype has high risk of lung cancer than wild type allele carriers, statistically significant interaction was found between genotype and mRNA [69]. In a recent leukemia study, *OGG12* has been found downregulated as it is key factor of cancer cells that DNA repair systems are deregulated hence decreased expression of the gene [70]. In present study, expression of *OGG12* is found to be downregulated in glioma tissues compared to controls thus, strongly in accordance with previously published results [71].

*PARP1* senses the nicks in single stranded and double stranded DNA breaks and seals them caused by oxidative stress. It is found to be hyperactive in oxidative stress and this hyperactivation also leads to cell death [72]. The stressful conditions like inflammation excitotoxicity, ischemia, and oxidative stress cause over activation of *PARP1* [73]. In previous studies it has been found to be overexpressed in various tumors like breast, ovary, skin, brain and lungs and this upregulation is linked to the enhanced anti-apoptotic property of tumors [74]. It has also found overexpressed in NSCLC [75], prostate [76], colorectal cancer [77] and gliomas [74]. Overexpression of *PARP1* is linked to poor survival overall in cancer [74]. In current study the expression of *PARP1* was also found over expressed. *PARP1* is found to be linked with the mechanisms relating to proapoptotic mechanisms [76] and overexpression might be due to defective cleavage of *PARP1* and reduced apoptosis hence long survival of cancer cells [33].

*HIF1α* is main regulator of hypoxia, cell cycle, metastasis, angiogenesis, metabolism and proliferation [22, 78]. Over expression of *HIF1α* has been observed in different cancers such as gastric [79] pancreatic cancer [80], NSCLC [81], thyroid cancer [82] and leukemia [83]. In current study *HIF1α* was found to be upregulated in glioma as compared to the controls supporting the previous results. Chen *et al.*, (2020) has reported that upregulation of *HIF1α* was found to be associated with activation of other oncogenic pathways like anti-tumor immunosuppression, cell cycle, EMT etc. This association has showed tumor promoter role of HIF1α [22]. HIF1α is stabilized with high ROS levels and provides advantage to tumor cells in division [78].

GDH plays important role in glutamine metabolism and its role in tumor cells is of great interest. GDH plays key role in glutathione which is important for maintaining redox balance in cell [84]. In this study GDH is found to be downregulated in glioma compared to healthy control samples. Deregulation of GDH has been observed in several cancers like thyroid cancer [85], head and neck cancer [86], breast cancer [87], lung cancer [21] and gliomas [88]. It has been found downregulated in lung cancer patients [89]. Expression level of mitochondrial sirtuins was correlated with selected related genes and significant correlation was observed between mitochondrial sirtuins and selected genes. This correlation results showed the combine involvement of selected gene in increased risk of glioma. Diagnostic and prognostic values of selected genes were also measured in present study. ROC curve analysis was performed and >80 percent diagnostic sensitivity was measured in case of SIRT3 and SIRT5 genes in glioma patients. In case of prognostic values of selected genes Kaplan Meier analysis was performed and significant deregulated expression of mitochondrial sirtuins was found linked with the decreased survival rate in glioma patients. Tan et al., (2020) has also reported that deregulation of mitochondrial sirtuins has found associated with decrease survival and increased hazard ratio of lower grade glioma [90]. Previous study has reported the downregulation of SIRT5 gene in glioblastoma and this deregulation was found associated with poor prognosis [91]. Prognostic values of mitochondrial sirtuins have been reported in different cancers such as breast cancer, colon cancer and non- small cell lung carcinoma [92]. Fu et al., (2020) has reported the prognostic value of mitochondrial situins in prostate cancer [93].

In present study DNA damage was measured by single cell gel electrophoresis and significant high number of comets were observed in glioma patients compared to the controls.

Along with this, some other parameters like %DNA in tail, olive tail moment and tail length were also analyzed, and they were found significantly high in glioma samples compared to healthy samples. DNA damage is any physical or chemical change in DNA in cells, in its result interpretation and transmission of genetic information is affected. The damage can be done by multiple factors like chemicals, radiations free radicles and topological changes which ultimately all result in different forms of damage. ROS is well recognized DNA damage mediator [94]. In present study, oxidative stress was also assessed by measuring the antioxidant level and significant increased oxidative stress was observed. ROS also induces DNA damage by oxidizing nucleoside bases (8-oxo guanine) which can lead to G-A and G-T transversions if unrepaired. ROS accumulation also results in mitochondrial DNA lesions, strand breaks and degradation of mitochondrial DNA [95].

Metabolic assays like ATP, NAD+ and glutathione level determination were also performed. In gliomas metabolic pathways like glycolysis, lipid and amino acid metabolism are found to be abnormally regulated. As ATP is the core of these pathways it has inevitably significant impact on glioma behaviors [96]. ATP is found to be at low levels in extracellular fluids in healthy tissues. While in case of some diseases including cancer, ATP level is increased [97], like in gastric cancer [98]. In our study level of ATP was also found high compared to the healthy controls.

Nicotinamide adenine dinucleotide (NAD) is a co-enzyme that actively involved in regulation of metabolic pathway. These pathways are serine biosynthesis, glycolysis and tricarboxylic acid (TCA) cycle [99, 100]. As NAD is restored so frequently and act as fuel for tumor cell, promote the uncontrolled cell division, proliferation, and survival. Elevated levels of NAD enhance glycolysis by the GAPDH and LDH, as they require NAD as coenzyme. NAD also serves as substrate for sirtuins and PARP hence mediate deacetylation and poly-ADP-ribosylation [101]. Intracellular NAD in mammals is controlled by two enzymes nicotinamide phosphoribosyltransferase (NAMPT) and nicotineamide mononucleotide adenyltransferase (Nmnat) [100]. In mammals Nmnat have three isozymes and they have different subcellular locations. NMNAT1 in nucleus, Nmnat2 in golgi apparatus, NMNAT3 in mitochondria and NAMPT is located in cytoplasm [102, 103]. In mitochondria, NAD utilization is in TCA cycle, oxidative phosphorylation, and Fatty acid oxidation [104]. In our study NAMPT, NMNAT1 and NMNAT3 were found significantly upregulated in glioma samples compared to the controls. Song et al., (2013) has reported that NMNAT1 is found located on chromosomal region which gets deleted frequently in cancers such as in lung cancer, ultimately resulting in advantage to tumor development [105]. Limited number of studies have been reported for NMNAT3. Anderson et al., (2017) has reported increased expression of NMNAT3, NAD level and enhanced energy level in mouse model [106]. NAMPT has been reported upregulated in different cancers like, colorectal cancer [107] prostate, breast and gastric cancers and this dysregulation is linked to promote tumor glycolysis, metastasis, invasion, proliferation, survival, and chemotherapy resistance [108].

## Conclusion

The present study demonstrated the expression variation of selected genes in glioma patients. The deregulation of these genes along with other metabolism deregulation resulted in increased oxidative stress hence resulting in unrepairable DNA damage and reduced survival in glioma patients. This study also provides potent insights into use of mitochondrial sirtuins and related genes as diagnostic and prognostic biomarkers. Several limitations are needed to be considered in present study such as study should incorporate oxygen consumption rate of glioma patients using seahorse analysis for better understanding of role of mitochondrial

abnormalities in glioma patients. Our study size is small, further validation studies with large sample size should be done to illuminate the mechanistic role of selected gene in cancerogenesis of different region including glioma.

## Acknowledgments

All authors would like to acknowledge the patients and normal individuals who contributed to this research work: we also acknowledge Hospital staff (Pakistan institute of medical sciences Islamabad (PIMS) for their cooperation.

## Author Contributions

**Conceptualization:** Ishrat Mahjabeen.

**Data curation:** Maria Fazal Ul Haq, Muhammad Zahid Hussain, Zertashia Akram, Sumaira Fida Abbasi, Mahmood Akhtar Kayani.

**Formal analysis:** Maria Fazal Ul Haq, Rabia Shafique.

**Investigation:** Maria Fazal Ul Haq, Muhammad Zahid Hussain, Nadia Saeed, Rabia Shafique, Sumaira Fida Abbasi.

**Methodology:** Maria Fazal Ul Haq, Zertashia Akram, Nadia Saeed, Rabia Shafique, Sumaira Fida Abbasi.

**Project administration:** Muhammad Zahid Hussain, Sumaira Fida Abbasi.

**Software:** Maria Fazal Ul Haq, Muhammad Zahid Hussain, Zertashia Akram, Nadia Saeed, Rabia Shafique, Mahmood Akhtar Kayani.

**Supervision:** Mahmood Akhtar Kayani.

**Validation:** Ishrat Mahjabeen, Mahmood Akhtar Kayani.

**Visualization:** Ishrat Mahjabeen, Mahmood Akhtar Kayani.

**Writing – original draft:** Ishrat Mahjabeen, Mahmood Akhtar Kayani.

**Writing – review & editing:** Ishrat Mahjabeen, Mahmood Akhtar Kayani.

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
