## [Decision Letter · Decision Letter 0]

24 Oct 2022

PONE-D-22-26011Oncometabolic Role of Mitochondrial Sirtuins in Glioma Patients.PLOS ONE

Dear Dr. Mahjabeen,

Thank you for submitting your manuscript to PLOS ONE. After careful consideration, we feel that it has merit but does not fully meet PLOS ONE’s publication criteria as it currently stands. Therefore, we invite you to submit a revised version of the manuscript that addresses the points raised during the review process.

We look forward to receiving your revised manuscript.

Kind regards,

Erika Di Zazzo

Academic Editor

PLOS ONE

Journal Requirements:

https://www.futuremedicine.com/doi/10.2217/fon-2021-0264

https://www.mdpi.com/2072-6694/9/7/90/htm

https://www.futuremedicine.com/doi/10.2217/fon-2019-0365

https://www.frontiersin.org/articles/10.3389/fonc.2018.00622/full

In your revision ensure you cite all your sources (including your own works), and quote or rephrase any duplicated text outside the methods section. Further consideration is dependent on these concerns being addressed.

"This study was supported by the grants from higher education commission of Pakistan as well as the COMSATS institute of information Technology Islamabad."

Reviewers' comments:

Reviewer's Responses to Questions

**Comments to the Author**

1. Is the manuscript technically sound, and do the data support the conclusions?

Reviewer #1: Yes

Reviewer #2: Yes

2. Has the statistical analysis been performed appropriately and rigorously? 

Reviewer #1: Yes

Reviewer #2: Yes

3. Have the authors made all data underlying the findings in their manuscript fully available?

Reviewer #1: Yes

Reviewer #2: Yes

4. Is the manuscript presented in an intelligible fashion and written in standard English?

Reviewer #1: Yes

Reviewer #2: Yes

5. Review Comments to the Author

Reviewer #1: Comments to Authors:

This is an interesting work PONE-D-22-26011 Oncometabolic Role of Mitochondrial Sirtuins in Glioma Patients. which aims to clarify the regulation of mitochondrial sirtuins and related genes in glioma cells. This work lays the foundation for future research in the study of gliomas. To improve your manuscript, some suggestions are listed below:

1) The abstract is quite long and you may risk losing the reader’s attention. I suggest you rewrite it in a more synthetic way in order to just highlight the contents.

2) The introduction turns out to be not very fluid: we appreciate the meticulous characterization of all the studied genes, but we invite you to improve the fluidity of this part.

3) As mentioned in the introduction, mitochondria are fundamental organelles involved in many biological processes like ATP production and apoptosis.

In glioma cells, the necroptosis process is also present and actively studied. We strongly suggest you read and cite it in your work (DOI: 10.1038/s41420-022-00974-x )

4) We encourage you to double check your manuscript font as some variances were found. We also ask you to specify why OGG12 is sometimes written as OGG1-2α.

5) The resolution of table 2 is too low. It results blurred and pixelated. Please, check and improve the quality of this image.

6) All your figures require major improvements and they are all presented in different sizes, making them visually chaotic.

6.1) in Figure 1, your graphs are not uniform. Some histograms have bold outlines while other have not. Also make sure your titles on the y axis have all the same character size.

6.2) In Figure 2, figure 3F, figure 5 and figure 6 are too small and difficult to read. We ask you to make them bigger and clearer for the reader.

6.3) Similarly, figures 7 and 8 have small tables under the graphs. Please, improve their resolution and size.

7) We recommend you to to standardize the letters indicating the various images of the panels. Some are bigger, while other are smaller and they need to be aligned in order to make your panel cleaner. Furthermore, we advise you to choose the same square shape for all the letters.

Reviewer #2: The authors have submitted a ms aimed at demonstrating deregulation of several genes resulting in

increased oxidative stress and DNA damage related to worse survival of patients with glioma

The authors also provide evidence that mitochondrial sirtuins and sirtuin-related genes can play

a role of diagnostic and prognostic biomarker for this neoplasm

The ms is clear and the discussion consistent with the results

However the authors should provide more details on some aspects of their work:

Can epileptic patients be considered "healthy controls" with no abnormality affecting the deregulation of genes and the pathways the authors show to be involved ? In which occasion and why the samples have been obtained from the epileptic patients ?

Why the authors di not carry on any seahorse analysis to achieve a better understanding of the metabolic abnormalities ?

6. PLOS authors have the option to publish the peer review history of their article (what does this mean?). If published, this will include your full peer review and any attached files.

Reviewer #1: No

Reviewer #2: **Yes: **Luciano Mutti

---

## [Author Response · Author response to Decision Letter 0]

22 Dec 2022

Journal Requirements:

We go through the manuscript and arrange according to Plos One style requirement as per suggestion of the worthy editor.

https://www.futuremedicine.com/doi/10.2217/fon-2021-0264

https://www.mdpi.com/2072-6694/9/7/90/htm

https://www.futuremedicine.com/doi/10.2217/fon-2019-0365

https://www.frontiersin.org/articles/10.3389/fonc.2018.00622/full

In your revision ensure you cite all your sources (including your own works), and quote or rephrase any duplicated text outside the methods section. Further consideration is dependent on these concerns being addressed.

Overlapping texts have been rephrased as per suggestion of the worthy editor.

Additional information regarding the participant consent and required detail of ethical review board have been added and highlighted red in Materials and methods section as per suggestion of the worthy editor.

"This study was supported by the grants from higher education commission of Pakistan as well as the COMSATS institute of information Technology Islamabad."

Funding information has been corrected and removed from the manuscript as per suggestion of the worthy editor. 

Correct funding statement has been added in cover letter as per suggestion of the worthy editor.

ORCID information has been added as per suggestion of the worthy editor.

Reviewer #1: 

1) The abstract is quite long and you may risk losing the reader’s attention. I suggest you rewrite it in a more synthetic way in order to just highlight the contents.

Abstract has been rewritten in more comprehended form (highlighted red) as per suggestion of the worthy review.

2) The introduction turns out to be not very fluid: we appreciate the meticulous characterization of all the studied genes, but we invite you to improve the fluidity of this part.

Introduction section has been thoroughly checked/corrected (highlighted red) and fluidity of the introduction section has been improved as per suggestion of the worthy reviewer.

3) As mentioned in the introduction, mitochondria are fundamental organelles involved in many biological processes like ATP production and apoptosis.

In glioma cells, the necroptosis process is also present and actively studied. We strongly suggest you read and cite it in your work (DOI: 10.1038/s41420-022-00974-x )

Suggested information has been added in introduction section (highlighted red) as per suggestion of the worthy reviewer.

4) We encourage you to double check your manuscript font as some variances were found. We also ask you to specify why OGG12 is sometimes written as OGG1-2α.

Manuscript has been thoroughly checked and correct term (OGG1-2α; mitochondrial subunit of OGG1)) has been added throughout the manuscript as per suggestion of the worthy reviewer.

5) The resolution of table 2 is too low. It results blurred and pixelated. Please, check and improve the quality of this image.

Quality and pixel of Table 2 has been improved for more clarification as per suggestion of the worthy reviewer.

6) All your figures require major improvements and they are all presented in different sizes, making them visually chaotic.

All suggested changes have been made in figures as per suggestion of the worthy reviewer.

6.1) in Figure 1, your graphs are not uniform. Some histograms have bold outlines while other have not. Also make sure your titles on the y axis have all the same character size.

Figure 1 has been improved and all histograms are made uniform with respect to outlines. Same size character has been used in titles of Y axis as per suggestion of the worthy reviewer.

6.2) In Figure 2, figure 3F, figure 5 and figure 6 are too small and difficult to read. We ask you to make them bigger and clearer for the reader.

Figure 2, Figure 3F, Figure 5 and Figure 6 have been improved with respect to size and clarity as per suggestion of the worthy reviewer.

6.3) Similarly, figures 7 and 8 have small tables under the graphs. Please, improve their resolution and size.

Tables associated with Figures 7 and 8 have been improved with respect to size and resolution as per suggestion of the worthy reviewer.

7) We recommend you to to standardize the letters indicating the various images of the panels. Some are bigger, while other are smaller and they need to be aligned in order to make your panel cleaner. Furthermore, we advise you to choose the same square shape for all the letters.

All images have been improved and presented in more uniform way as per suggestion of the worthy reviewer. Figures have been removed from the main manuscript file and uploaded separately as till files as per instruction of the worthy editor.

Reviewer #2: 

1. The authors have submitted a ms aimed at demonstrating deregulation of several genes resulting in increased oxidative stress and DNA damage related to worse survival of patients with glioma. The authors also provide evidence that mitochondrial sirtuins and sirtuin-related genes can play a role of diagnostic and prognostic biomarker for this neoplasm.

Previous studies reporting the diagnostic/prognostic values of mitochondrial sirtuins in brain tumor and other cancers, has been added in discussion section (highlighted red) as per suggestion of the worthy reviewer.

2. However the authors should provide more details on some aspects of their work:

Can epileptic patients be considered "healthy controls" with no abnormality affecting the deregulation of genes and the pathways the authors show to be involved ? In which occasion and why the samples have been obtained from the epileptic patients ?

Since brain tumor surgery is restricted to tumor section only and the surgeon attempts to remove the tumor without damaging vital brain tissue. The surgeon may remove as much of the tumor as possible without taking any distant region. Due to these restrictions, we could not take the adjacent normal tissue sections to be used as controls like other cancers. Controls were taken from the brain surgeries of epilepsy patients or from other surgeries of brain. 

Previous studies have reported that epileptic patients’ sections can be taken/used as controls (doi: 10.1016/j.cell.2020.05.007: doi: 10.4049/jimmunol.1301966.).

Our results have also shown the significant difference between the expression pattern of selected genes/pathway in glioma patients and epileptic patients section taken as controls. 

3. Why the authors did not carry on any seahorse analysis to achieve a better understanding of the metabolic abnormalities ?

Oxygen consumption rate is measured by most advanced technology of seahorse based bioenergentic analysis. We were unable to perform this analysis due to unavailability of seahorse analyzer and added (highlighted red) this technique in our study limitations in discussion section. However, measurement of ATP and NAD has been performed which is considered a basic step for the measurement of metabolic rate in cell.

---

## [Decision Letter · Decision Letter 1]

1 Feb 2023

Oncometabolic Role of Mitochondrial Sirtuins in Glioma Patients.

PONE-D-22-26011R1

Dear Dr. Ishrat Mahjabeen,

We’re pleased to inform you that your manuscript has been judged scientifically suitable for publication and will be formally accepted for publication once it meets all outstanding technical requirements.

Kind regards,

Erika Di Zazzo

Academic Editor

PLOS ONE

Additional Editor Comments (optional):

Reviewers' comments:

Reviewer's Responses to Questions

**Comments to the Author**

1. If the authors have adequately addressed your comments raised in a previous round of review and you feel that this manuscript is now acceptable for publication, you may indicate that here to bypass the “Comments to the Author” section, enter your conflict of interest statement in the “Confidential to Editor” section, and submit your "Accept" recommendation.

Reviewer #1: All comments have been addressed

2. Is the manuscript technically sound, and do the data support the conclusions?

Reviewer #1: Yes

3. Has the statistical analysis been performed appropriately and rigorously? 

Reviewer #1: Yes

4. Have the authors made all data underlying the findings in their manuscript fully available?

Reviewer #1: Yes

5. Is the manuscript presented in an intelligible fashion and written in standard English?

Reviewer #1: Yes

6. Review Comments to the Author

Reviewer #1: The authors have provided the manuscript with all the suggestions requested. All the comments were made in order to improve the fluidity of the text and to achieve images with higher resolutions.

7. PLOS authors have the option to publish the peer review history of their article (what does this mean?). If published, this will include your full peer review and any attached files.

Reviewer #1: No

---

## [Editor Report · Acceptance letter]

9 Feb 2023

PONE-D-22-26011R1 

Oncometabolic Role of Mitochondrial Sirtuins in Glioma Patients 

Dear Dr. Mahjabeen:

I'm pleased to inform you that your manuscript has been deemed suitable for publication in PLOS ONE. Congratulations! Your manuscript is now with our production department. 

Kind regards, 

on behalf of

Dr. Erika Di Zazzo 

Academic Editor

PLOS ONE